# “I’ll Be There”: Informal and Formal Support Systems and Mothers’ Psychological Distress during NICU Hospitalization

**DOI:** 10.3390/children9121958

**Published:** 2022-12-13

**Authors:** Miri Kestler-Peleg, Varda Stenger, Osnat Lavenda, Haya Bendett, Shanee Alhalel-Lederman, Ayala Maayan-Metzger, Tzipora Strauss

**Affiliations:** 1School of Social Work, Ariel University, Ariel 40700, Israel; 2Department of Neonatology, Safra Children’s Hospital, Sheba Medical Center, Ramat Gan 52621, Israel

**Keywords:** preterm birth, neonatal intensive care unit, family-centered care, intolerance to uncertainty, medical staff support, social support, spousal support

## Abstract

Mothers of infants hospitalized in the Neonatal Intensive Care Unit (NICU) are at a high risk for psychological distress, which is of concern to health and social professionals due to the negative implications for mothers and infants. A model for explaining maternal psychological distress, consisting of intolerance to uncertainty and support from informal (spouse, family, and friends) and formal (medical staff) systems was examined. Data was collected from one of the largest NICUs in Israel; 129 mothers of 215 preterm infants completed self-report questionnaires regarding their background variables, intolerance to uncertainty, perceived informal support and perceived medical staff support. The NICU’s medical staff provided indicators for the infants of participating mothers. A hierarchical multiple regression analysis was conducted. The examined model explained 29.2% of the variance in maternal psychological distress. Intolerance of uncertainty positively predicted psychological distress. Informal support, and in particular, spousal support negatively predicted psychological distress above and beyond intolerance of uncertainty. Medical staff support negatively predicted psychological distress above and beyond intolerance to uncertainty and informal support. Our findings suggest that maternal psychological distress is reduced through a family-centered care approach in NICUs. Medical professionals and social services should develop further solutions for addressing preterm mothers’ need for certainty and support.

## 1. Introduction

Preterm birth is a stressful event [1] that is associated with increased maternal psychological distress [2] and decreased maternal wellbeing [3]. In most cases, it is unexpected and often occurs in an emergency atmosphere [2]. Parents, especially mothers, of infants admitted to the Neonatal Intensive Care Unit (NICU) can feel powerless and helpless [4,5]. Therefore, they may be more stressed and vulnerable to emotional difficulties than mothers of full-term infants [6]. The current study focuses on the role of social support (both informal social support, i.e., spouse, family, and friends, and formal social support, i.e., the NICU’s medical staff) in predicting mothers’ psychological distress during NICU hospitalization, while considering mothers’ intolerance to the uncertainty that this situation generates.

### 1.1. Mothers’ Psychological Distress and Uncertainty during NICU Hospitalization

Approximately 5% to 18% of live births worldwide are preterm births, i.e., infants born after a gestation period of less than 37 weeks [7]. The comprehensive hormonal and emotional preparation that mothers undergo with the birth of a full-term infant is partly lacking with a preterm birth [8,9]. A sharp, sudden, and uncontrolled separation of the infant from the mother occurs during delivery [10]. As such, a potentially happy occasion becomes a situation of crisis, accompanied by reactions of stress, shock, trauma, outbursts of crying, anger, feelings of guilt and anxiety [4]. The infant is usually placed in an incubator in the NICU immediately after delivery and receives artificial respiration and intravenous nutrition as needed [9]. The mothers, who were anticipating a healthy infant, find themselves contemplating an infant that does not meet their expectations, does not look as they hoped and is not yet completely formed [1]. The infant appears fragile, attached to various kinds of medical equipment, and its life is in danger. The infant is also separated from the mother while in the incubator, and their care is out of the mother’s control [5]. Mothers who are challenged by many changes, as is the case with prematurity [1,11], and uncertainty [10] are also at increased risk for developing psychological distress, which may be expressed as depression, anxiety, and post-traumatic stress symptoms [12].

Psychological distress is an umbrella term that describes a variety of mental difficulties. It is an emotional experience of discomfort that results in harm to the individual [13]. The distressing nature of the NICU environment for mothers of preterm infants encompasses a physical environment characterized by bright lights, chemical odors, and the noise caused by medical equipment–life support and monitoring. Most disturbing, however, is seeing their own newborn infant in the NICU environment, with all its medical equipment and personnel [14]. Furthermore, the disruption of the parental role with all its expectations as well as feelings of disappointment and frustration regarding anticipated parenting tasks that are denied to them may result in considerable distress [15]. Hospitalization of the infant may also lead to maternal mood swings that alternate between hope and hopelessness [11]. This difficult situation leads to great uncertainty [16]. While the experience of uncertainty is pervasive in human existence [17], the NICU represents a huge source of ambiguity regarding the maternal role, the infant’s prognosis, and the communication between parents and medical personnel [10].

The difficulty of dealing with uncertainty is reflected by intolerance to uncertainty. This is the individual’s difficulty in tolerating the discomfort caused by uncertainty, viewing uncertain future events as threatening, undesirable, and a source of negative consequences [18]. In an attempt to impose control or evade uncertainty, individuals who find it difficult to cope with uncertainty often resort to negative estimations of their own coping abilities and experience high levels of psychological distress [17,19]. Moreover, mothering a preterm infant that is hospitalized in the NICU is characterized by high levels of uncertainty regarding the infant’s duration of hospitalization, prospects of survival and the potential outcomes of preterm birth, as well as the invasiveness of infant hospitalization in terms of both its immediate and long-term implications [11]. As such, the experience of having a preterm infant hospitalized in the NICU can elevate the mother’s levels of psychological distress [16].

Nevertheless, factors that have the potential to reduce the level of psychological distress among mothers of hospitalized NICU infants are understudied. However, research on distress-related mechanisms has shown that social systems play a crucial role as protective and facilitative factors [20]. Therefore, the present study focuses on social support systems as a positive aspect that is known to promote wellbeing [20,21].

### 1.2. Social Support

The present study follows the salutogenic model for health [22] to further explore the role of social support systems [20,21] in predicting mothers’ psychological distress in the face of the stressful occasion of preterm birth and subsequent NICU hospitalization. A thematic analysis confirmed that social support is a crucial resistance resource, i.e., it facilitates the individual’s ability to successfully cope with various sources of stress [23].

Social support can be defined as that which assists individuals in believing the following about themselves: they are cared for, loved, and valued as part of a network connected by mutual obligations [24]. It is a general term that embraces various types of assistance, including emotional, instrumental, informational, etc., which are gained from relationships with significant others [25]. The importance of support systems as coping resources for dealing with stressful life events is well documented [26,27], particularly in the context of the NICU [3]. Based on the salutogenic model for health [20], our main assumption is that social support is a crucial resistance resource when mothers are dealing with the great uncertainty of a preterm birth and subsequent infant NICU hospitalization. As such, social support will predict reduced maternal psychological distress.

Social support is often considered and tested as a generic construct [28,29], despite its varied sources. However, social support may encompass both informal (spousal, family and friends) and formal ( organizations) components. The current study focuses on the contribution of several sources of social support to the prediction of psychological distress in mothers during NICU hospitalization.

### 1.3. Informal Social Support

Informal social support comes from close relatives and from people who, without depending on formal organizations, offer their help informally in a way that is not regulated or institutionalized [30]. Apparently, informal social support is the greatest source of social support for coping with stress [31].

Spousal support: The main support during NICU hospitalization is the mother’s spouse or partner [20,32], which is usually the infant’s father. Fath5rs of preterm infants, who are expected to provide support to the mothers, also experience shock and stress following an unpredictable preterm birth and the challenging conditions of NICU hospitalization [33]. Indeed, spousal support was found to be a predictor of reduced levels of mothers’ stress during NICU hospitalization [3].

Family support: Family support was found to predict a decrease in depressive symptoms among preterm mothers [34]. At the same time, the extended family itself has to deal with uncertainty regarding the health of the new infant in addition to the strict policies of the NICU regarding the care and contact of family members. The role of the grandparents among the extended family is particularly significant [35], and the involvement of the parents’ parents is appreciable for preterm mothers.

Friend support: Compared to other support resources, friend support is relatively understudied and has rarely been researched among mothers [36]. However, the option of NICU mothers to be actively supported by friends is limited due to NICU policy and the firm restrictions regarding visits of friends and their participation in the care of the hospitalized infant. Nevertheless, friends’ emotional support and assistance at home and in other circumstances outside the hospital are still significant. Although a negative association was found between friend support and psychological distress [37,38], to the best of our knowledge, no study has examined the unique contribution of friend support to the prediction of psychological distress in preterm mothers during the hospitalization period.

### 1.4. Formal Support

Formal social support is the assistance that is provided either through an organized group or an agency [39]. Most formal support for preterm mothers is provided through the medical staff. Awareness regarding the significance of support from medical staff for NICU mothers has increased over the past decade [40,41].

Models of family-centered care (FCC) have emerged as optimal in NICUs worldwide due to the recognition of the need for medical knowledge regarding maternal psychological states and for practical support for improved physical conditions [10]. In the FCC, clinician and family engagement is informed by mutual respect and shared decision-making, generating psychosocial support and allowing for the allocation of hospital resources to attain the goal of family well-being and involvement [14]. Indeed, a recent systematic review indicated that the FCC approach contributes to improvements in mothers’ mental health [42]. As such, the FCC model is the basis of the approach taken by the NICU where the current study was conducted. As part of this FCC implementation, for example, a text message is sent every morning to the parents, informing them of the infant’s current condition and what transpired over the previous hours. Peer groups are organized for the parents as well as for the siblings and grandparents of the infant, while family members are allowed to participate in the care of the preterm infants.

### 1.5. The Present Study

The goal of the present study is to examine the predictive power of social support, both formal and informal, of maternal psychological distress above and beyond mothers’ ability to tolerate uncertainty as a result of preterm birth and NICU hospitalization. In particular, we examined the contribution of the social support provided by NICU medical staff to mothers’ psychological distress after preterm birth.

According to the literature review, we hypothesized the following:

**H1:** *Mothers’ intolerance of uncertainty will positively predict maternal psychological distress*.

**H2:** *Mothers’ social support, provided by informal sources, will negatively predict maternal psychological distress above and beyond their intolerance of uncertainty*.

**H3:** *Mothers’ social support provided by the NICU’s medical staff will negatively predict maternal psychological distress above and beyond the support provided by informal sources*.

## 2. Methods

### 2.1. Study Procedure and Sample

Data were collected from March 2017 until May 2018 at the NICU in one of Israel’s largest medical centers, where self-report questionnaires were completed by 129 Israeli mothers of 215 preterm infants. Ethical approval for conducting the study was granted by the medical center’s Helsinki committee (No. 2904-16-SMC, granted on 2 February 2016 and extended again on 4 February 2019). A NICU social worker reached out to hospitalized mothers 2–3 days before their release day, asking for their consent to participate in the research. Those mothers who agreed were requested to sign a consent form and were handed a paper copy of a self-report questionnaire to complete. It was explained to them that their hospital service or NICU treatment would in no way be affected by their agreement or lack thereof to participate in the study. In the following days, the social worker returned to collect the forms. At the end of the data collection process, a researcher recorded the participants’ responses using IBM SPSS Statistics.

The ages of the mothers ranged from 21 to 48 years old (*M* = 34.2, *SD* = 5.4), with an average education of 15.6 years (*SD* = 2.7). Most of the sample reported living with a partner (94.2%) for an average of 7.7 years (*SD* = 4.8) and had given birth to an average of 2.4 children (*SD* = 1.6). Most of the participants reported working full-time before delivery (72.1%) with an average Israeli income or higher (17.1% = income levels lower than average, 43.4% = average income, and 43.1% = higher than average income). The infants had gestational ages ranging between 24 and 37 weeks (*M* = 31.4, *SD* = 2.7), with birth weights between 390 and 2930 gr’ (*M* = 1564.2, *SD* = 462.1).

### 2.2. Measures

All measures used in the present study are validated measures that are well known and widely used among researchers in the social sciences. Cronbach’s alpha index was added to each of the following descriptions to indicate the reliability of each measure.

*1. Psychological Distress Questionnaire:* This questionnaire was developed by Kessler et al. (2003) [43] and is used frequently [44,45]. Participants were asked to rate the extent to which they experienced the feelings described in each of the 6 items. Cronbach’s alpha calculated in the present study was 0.759.

*2. Intolerance of Uncertainty Questionnaire:* The Intolerance of Uncertainty Scale (IUS) [46] (Freeston et al., 1994) was used in the present study in its abbreviated version [47], which includes 12 items (IUS-12). The scale was generated to evaluate emotional, cognitive, and behavioral reactions to uncertainty encountered in regular life. Responses are given on a 5-point Likert scale. A higher score indicates that the participant struggled increasingly with uncertainty. This questionnaire has been validated worldwide [16,48]. In the present study, Cronbach’s alpha was 0.926.

*3.Social Support Questionnaire:* This 9-item scale is based on the theoretical conceptualizations of House (1981) and was developed into a questionnaire in Hebrew by Westman et al. (2004) [49]. It refers to received support, understanding, advice and assistance from spouse, family, and friends (three items per source of support). Participants were asked to rate how well the items described their relationships with their spouse, family, and friends on a 5-point Likert scale. In the present study, Cronbach’s alpha was 0.861 for spousal support, 0.913 for family support, and 0.922 for friend support.

*4. Medical Staff Support Questionnaire:* Based on a questionnaire developed by Bryanton et al. (1994) [50], this 16-item scale addresses various supportive medical staff behaviors, such as emotional support (reassuring and providing a sense of confidence), informational support (providing information, advice and/or giving feedback), and tangible support (providing direct aid, e.g., taking care of someone physically). Responses were given on a 5-point Likert scale. In the present study, Cronbach’s alpha was 0.939.

*5. Background variables:* The NICU’s medical staff provided information regarding the preterm infants (weight and gestational age), and mothers provided their background information (age, education, marital status, employment status and income).

## 3. Results

To test the study’s hypotheses, a hierarchical multiple regression analysis was conducted. Little’s Missing Completely at Random (MCAR) test was conducted to eliminate possible data bias due to missing data. The results confirmed that data were missing completely at random (*χ*^2^ (6) = 7.812, *p* = 0.252) and therefore did not impact the power of the statistical analyses. Table 1 presents the descriptive statistics of the study variables.

First, intolerance of uncertainty was entered in the regression. Next, various types of informal support (spousal, family, and friend support) were entered to assess their ability to predict maternal psychological stress above and beyond intolerance of uncertainty. Finally, formal support (NICU medical staff support) was entered in the third step of the regression to assess its prediction of maternal psychological stress, above and beyond intolerance of uncertainty and informal support. Table 2 summarizes the results of the hierarchical multiple regression analysis.

As indicated in Table 2, the overall model explained 29.2% of the variance in the maternal psychological distress. Consistent with H1, approximately 20% of the variance was accounted for by intolerance of uncertainty. An additional 7.8% of the explained variance was attributed to the various forms of informal support. This addition was found to be significant (*F*∆ (3125) = 4.464, *p* = 0.005), consistent with H2. Nevertheless, as indicated in the table, only spousal support was found to be a significant contributor to the explained variance in maternal psychological distress, and as predicted, it was found to be a negative predictor of distress (*β* = −0.296, *p* = 0.002; *β* = −0.282, *p* = 0.003). The addition of medical support provided by the NICU staff was found to be negatively significant (*β* = −0.216, *p* = 0.005) above and beyond the informal support sources (*F*∆ (1124) = 8.141, *p* = 0.005). The latter confirms H3.

## 4. Discussion

The aim of the present study was to examine the role of formal social support in predicting preterm maternal psychological distress in the context of great uncertainty compared with informal social support. The findings confirmed the study’s hypotheses by indicating that intolerance to uncertainty positively predicted psychological distress. Informal social support negatively predicted psychological distress above and beyond mothers’ intolerance of uncertainty. Spousal support negatively predicted maternal psychological distress. Finally, formal support, as reflected by medical staff support, negatively predicted preterm mothers’ psychological distress above and beyond intolerance to uncertainty and informal social support during NICU hospitalization.

### 4.1. Intolerance to Uncertainty

Of the three resources, intolerance of uncertainty was found to be the most powerful factor associated with mothers’ psychological distress. This finding is consistent with previous studies that examined the relationship between intolerance to uncertainty and psychological distress in other populations and stressful situations [51,52]. The difficulty of enduring uncertainty and the desire to clarify the uncertainty emerges here as a debilitating factor. The burden of the concerned mother who contends with the unknown (e.g., “Will the infant survive this difficult period?”, “What injuries will the infant suffer, in the short and long term?”, “What will the infant’s future look like?”) seems to be a crucial aspect of the maternal ability to cope with the stressful nature of having a preterm infant. The mother’s incapacity to deal with the uncertainty regarding the future is what restricts her resilience to maternal psychological distress. Despite its great relevance, to the best of our knowledge, intolerance to uncertainty has scarcely been examined thus far in the context of preterm birth [except for 34].

### 4.2. Social Support

In the last decade, the utilization of the salutogenic model has gained momentum in studies concerning postpartum adjustment, from an almost exclusive pathogenic position to an orientation that emphasizes the contribution of the environment to positive maternal outcomes [53]. In accordance with the salutogenic model, it appears that social support facilitates the abilities of preterm mothers to cope more successfully with stressors and contributes to the development of a sense of coherence [22]. Thus, social support emerged in this study as a resistance resource against psychological distress among mothers of preterm infants during NICU hospitalization. Consistent with previous studies that found that social support predicted lower levels of postpartum depression and emotional distress among mothers of preterm infants [54,55], the current study demonstrated that even though the circumstances of preterm birth and mothering a NICU hospitalized infant are threatening, upsetting and challenging, the more mothers perceive their surroundings (informal as well as formal) as supportive, the more their psychological distress is reduced.

### 4.3. Informal Support

The role of informal social support in predicting reduced stress and anxiety during the transition to motherhood is well recognized [37,56]. This contribution was validated in the present study as well, when informal support predicted decreased psychological distress among the sample of preterm mothers during NICU hospitalization. In the present study, after the significance of informal support systems was established, it was found that the key person for this support is the spouse. An essential component of the ability of the spouse, usually the father of the NICU-hospitalized infant, to be supportive is the nature of the dyadic relationship with the mother. These dyadic relationships place spouses in the closest position to preterm mothers, and accordingly, the essentiality of the support they offer. Support from the father, who also faces unexpected change, embodies a crucial resistance resource for maternal psychological distress. In such unique circumstances, if the spouse is perceived by the mother as supportive, it allows the mother to share the feelings that are caused by the NICU conditions as well as to consult, obtain comfort, and manage feelings of loneliness resulting from the duration of time spent in the hospital.

### 4.4. Formal Support

Formal support, which is reflected by medical staff support in the present study, negatively predicted mothers’ psychological distress during NICU hospitalization, above and beyond intolerance to uncertainty and informal support systems. This finding emphasizes the great importance of the FCC approach, in which medical teams provide assistance to worried mothers. A previous study indicated that NICU mothers want to participate in their infant’s care and crave honest, timely, comprehensive information regarding their infant’s health and potential outcomes [57]. In line with the salutogenic model [22,53], and to meet these expectations and improve the outcomes of preterm infants and their parents, the FCC model has been applied worldwide [58]. In accordance with the FCC model, the NICU where the current study was conducted has launched special projects that include, for instance, peer groups for parents, involving grandparents in the care of the infants, using technological innovations to involve parents in the treatment of the baby and improving efforts to communicate with the parents.

As such, the findings of the current study revealed that the medical staff’s support for mothers, according to the principles of FCC, plays a central role in their psychological state, even more so than the informal support that family or friends offer. This finding is consistent with previous studies that found that implementing the FCC model reduced parental psychological distress [57,58].

### 4.5. Study Limitations

Our model accounted for a substantial amount of the explained variance of maternal psychological distress, as well as involved medical staff support, which is a key factor for future recommendations and clinical practice implications. Even so, it is not without its limitations. Firstly, the data were only collected via self-report questionnaires. As such, the results may be biased as a result of social desirability and shared variance. Secondly, the measure of medical staff support referred to the medical staff alone; namely, doctors and nurses. Given the complexity of NICU admission, patients are treated by a multidisciplinary team, which was not considered in the present study. Thirdly, the sample lacks representativeness. The data were gathered in a sizeable public Israeli medical center. However, no information regarding potential participant refusal to participate was elicited. It may be that declining to participate in the study or not being invited to participate due to conditions deemed unstable and/or sensitive are indicators of different characteristics (e.g., skills for coping with high anxiety, lacking trust in others, or contending with an especially frightening infant medical condition), and this may have also biased the research findings. As such, these non-represented individuals may have responded differently, thereby altering the results in regard to their significance or direction. Finally, our cross-sectional study design does not permit us to draw any conclusions about directionality in terms of the results.

It is recommended that future research re-examine our associations by employing a longitudinal design. This could start with the fertility process and conclude two years following the birth, with several intervals of data collection. This type of research may result in superior identification of the pattern of changes in the predictors and enable positing the nature of the causality of the associations, linking them and the maternal psychological state. Additionally, generalization of the findings to other population segments requires conducting the same type of study in a range of cultures and countries. The importance of examining psychological distress in mothers and fathers of NICU-hospitalized preterm infants should be noted. Future studies should, therefore, focus on the role of the father as crucial and essential in childrearing.

### 4.6. Practical Implications

Some practical implications can be derived from our results, and these could be useful for NICU medical and social services in policy development and intervention design for mothers faced with preterm infant hospitalization, particularly mothers at risk of psychological distress. At the level of social services, policymakers ought to allocate greater clinical support to mothers of hospitalized infants. Practically-speaking, mothers can benefit from professional discussions concerning the fears surrounding the NICU experience, especially those outside the control of the mother. With the proper encouragement, mothers can thus develop patience and tolerance in the face of challenging daily circumstances.

In regard to spousal support, both the mother and father should be encouraged by professionals to play a role in caring for the hospitalized infant as a means of strengthening their spousal bond. This partnership is critical for the support system when confronted by a demanding situation in which loneliness and fear are also implicated.

The FCC approach and the development of additional programs that further address the support needs of mothers may increase the perception of the medical staff as a support and also enable the support of broader informal systems that include family members and friends. Finally, we find it is of utmost importance to have an open visitation policy that permits the involvement of fathers, family and friends in the infant’s care and activities [57]

## 5. Conclusions

The present study advances the understanding of psychological distress among mothers of preterm infants hospitalized in the NICU. Specifically, it demonstrates the significant role of the ability to bear uncertainty as well as the crucial role of spousal and medical staff support in preterm mothers’ psychological state. Since all of the study’s predictors can be influenced by targeted policies and professional interventions and in accordance with the salutogenic model, these findings contribute to the promotion of programs that aim to reduce the psychological distress of preterm mothers during NICU hospitalization.

## Figures and Tables

**Table 1 children-09-01958-t001:** Ranges, mean scores and standard deviations of the study variables.

	Minimum	Maximum	Mean	S.D.
Intolerance of uncertainty	1.17	5.00	2.892	0.899
Spousal support	1.00	5.00	4.351	0.755
Family support	1.00	5.00	3.736	1.018
Friend support	1.00	5.00	3.156	1.178
Medical staff support	1.80	5.00	4.278	0.614
Psychological distress	1.00	3.50	1.840	0.557

**Table 2 children-09-01958-t002:** Hierarchical regression coefficients for predicting maternal psychological distress by intolerance of uncertainty and formal and informal support.

	Step I	Step II	Step III
Intolerance of uncertainty	0.445 **	0.353 **	0.380 **
Spousal support		−0.296 **	−0.282 **
Family support		0.138	0.182
Friend support		−0.080	−0.090
Medical staff support			−0.216 **
Sig.	0.000	0.005	0.005
R^2^	0.198	0.275	0.292
R^2^ Change	0.198	0.078	0.045

** *p* < 0.01.

## Data Availability

Due to the nature of this research, the participants of this study did not agree to their data being shared publicly, so supporting data is not available.

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
