# Peer review of "“I’ll Be There”: Informal and Formal Support Systems and Mothers’ Psychological Distress during NICU Hospitalization"

_children, 2022, doi:10.3390/children9121958_

Round 1

Reviewer 1 Report

First of all, I would like to thank the authors for the opportunity to review the manuscript. I think there are a few aspects to review before the manuscript can be considered for publication.

-The topic seems to meet Children's criteria.

-The introduction is longer than I would expect for contextualization. 

-Giving birth to a preterm infant & Psychological distress and uncertainty during NICU hospitalization: can be merged and summarized as the second one is a result of the first statement. 

-Social support & Informal social support & Formal support: even do they are not the same, they are very much related so could potentially be merged and summarized. If so would help to reduce the introduction extension. 

-Line 214 - 129 mothers

-Measures used: I understand the Cronbach alpha is a way to prove they are valid for the sample even if are not validated in the present population. Please make this clear so any reader can see if is been previously validated or not. 

-Social support & Informal social support & Formal support: conversely to what I said before, makes sense here because is given by statistical differences. So well done here. 

-Limitations are well described, but I would point out the strength of medical staff support as a factor involved. This factor is important and key in future recommendations and clinical practice implications. 

-However, if the questionnaire treats medical staff as a multidisciplinary staff should be changed. I´ve tried to look for the questionnaire but couldn´t get it. Can we clarify if is only referred to as medical staff or is the general staff? Medical staff should be considered as a limitation due to the complexity of NICU admissions (other professionals involved). 

-Practical implications are mixed with the conclusion, please put them in separately and provide an independent, concise, and accurate conclusion. 

I would recommend correcting the introduction, reviewing a few minor mistakes and grammatical errors, connectors being repeated... (with proofreading should be enough), and having a 2nd review. 

Author Response

(Reviewer 1):

First of all, I would like to thank the authors for the opportunity to review the manuscript. I think there are a few aspects to review before the manuscript can be considered for publication. 

-The topic seems to meet Children's criteria.

 -The introduction is longer than I would expect for contextualization.

Response: Following the reviewer’s comment, we shortened the Introduction section from 2193 words to 1827 words (see: L.29-185).

 -Giving birth to a preterm infant & Psychological distress and uncertainty during NICU hospitalization: can be merged and summarized as the second one is a result of the first statement. 

Response: Following the reviewer’s comment, we merged the two subsections into one (see: L.42-98).

-Social support & Informal social support & Formal support: even do they are not the same, they are very much related so could potentially be merged and summarized. If so would help to reduce the introduction extension

Response: Following the reviewer’s comment, we merged the subsection of medical staff support with the section on informal support (see: L.152-171). Nevertheless, dividing the text in a way that distinguishes between social support, informal social support, and formal social support lays the foundation for the theoretical model examined in the current study. However, following the reviewer’s comment, we shortened the text in other ways (see: L.99-151).

-Line 214 - 129 mothers

Response: Thank you for this comment, following which we have corrected the typo (see: L.190). 

 -Measures used: I understand the Cronbach alpha is a way to prove they are valid for the sample even if are not validated in the present population. Please make this clear so any reader can see if is been previously validated or not. 

Response: Indeed, Cronbach’s Alpha is an index that verifies the quality of the measure. All measures used in the present study are popular and used widely among social sciences researchers. Following the reviewer’s comment, we have added this information to the Measures subsection (see: L.212-214).

 -Social support & Informal social support & Formal support: conversely to what I said before, makes sense here because is given by statistical differences. So well done here. 

 Response: Thank you.

-Limitations are well described, but I would point out the strength of medical staff support as a factor involved. This factor is important and key in future recommendations and clinical practice implications. 

Response: Following the reviewer’s comment, we pointed out the strength of medical staff support in the Limitations subsection (see: L.356-357).

-However, if the questionnaire treats medical staff as a multidisciplinary staff should be changed. I´ve tried to look for the questionnaire but couldn´t get it. Can we clarify if is only referred to as medical staff or is the general staff? Medical staff should be considered as a limitation due to the complexity of NICU admissions (other professionals involved). 

Response: The questionnaire refers to the medical staff, i.e., doctors and nurses. Therefore, following the reviewer’s comment this limitation was added to the Limitation subsection (see: L.360-362).

-Practical implications are mixed with the conclusion, please put them in separately and provide an independent, concise, and accurate conclusion. 

Response: Following the reviewer’s comment, we distinguished practical implications and conclusions, and split them into two separate subsections (see: L.403).

I would recommend correcting the introduction, reviewing a few minor mistakes and grammatical errors, connectors being repeated... (with proofreading should be enough), and having a 2nd review. 

Response: Thank you for this recommendation. Following your suggestion, we did one more proofreading and corrected wording mistakes, grammatical errors, and repeated connectors that we found in the manuscript (For example, see: L.8, 32, 103, 132).

Reviewer 2 Report

Thank you for the opportunity to review this paper which has the potential to inform practice in the area of supporting parents with premature babies in NICU. This paper aimed to understand maternal psychological distress of mothers with babies on a NICU and its relation to intolerance to uncertainty and formal and informal support. Results showed that formal support negatively predicted psychological distress above and beyond intolerance to uncertainty and informal social support.

Overall the paper is well structured and written with appropriate methodology. I have a few comments which I think will enhance the paper.

Abstract

1)      I would expect to see in the abstract or main body, the dates in which the research took place. Could this be added to either the abstract or methods sections.

Background

2)      This is very informative but very long, taking over a third of the overall paper. Would some of this be better placed in the discussion section or left out entirely. The first paragraph or two of each sections may be enough or a reduction in each section.

3)      Line 75-6 ‘The sight of their tiny infants connected to equipment by 75 tubes and wires and surrounded by medical personnel is also very disturbing’. There needs to be a reference.

4)      78-81 ‘Preterm mothers report feelings of 78 disappointment and frustration that they cannot perform normal parenting tasks (e.g., 79 holding, feeding, changing diapers)...’ There needs to be a reference.

Methods

5)      Methods section usually starts with an explanation of the method chosen and then goes onto the finding. This section seems to be in the wrong order.

The section on measures would be better described before providing the descriptive data, in a similar manner to your other publication or add a statement to refer to your other paper for a description of the methods. Kestler-Peleg, M., & Lavenda, O. (2022). Personal resources associated with peripartum depression among mothers of NICU 504 hospitalized preterm infants. Psychology & Health, 37, 712-730. https://doi.org/10.1080/08870446.2021.1873336

6)      There is not enough information on the method of data collection to reproduce this study. Adding the information below would aid understanding of the process of consent and data collection.

The time period or date of the study needs to be added.

More description around the process of giving out the questionnaires would help to understand the process. When were they given out, on admission to NICU? Withing a week of admission?

Was it a paper copy, self-completed?

7)      Line 214, this says ‘29 Israeli mothers’. The introduction says 129. Should this be 129?

8)      Line 223, the authors guidelines state the manuscript must contain ‘a statement including the project identification code, date of approval, and name of the ethics committee’. Please add these as required. 

Results

Appropriate statistical analysis for the screening tools used and hypothesis.

Discussion

The discussion clearly lays out the findings in relation to the hypothesis and includes limitations and practical implications.

References

9)      There are a large number of references that are over 5 years (the time period suggested by the journal). I understand the need for the use of seminal papers and references to screening tools which are used which may be decades old. There are other references which could be updated, for example lines 432, 473, 496, 498 the reference for mothers experience of giving birth to a premature infant. There are numerous studies looking at experiences of mothers with premature babies.

e.g. The forgotten mothers of extremely preterm babies: a qualitative study. Fowler et al. 2019 or

Experiences of mothers with preterm babies at a mother and baby unit of a tertiary hospital: a descriptive phenomenological study. Lomotey et al 2019

Could these references be reviewed and updated where possible.

Author Response

(Reviewer 2):

Thank you for the opportunity to review this paper which has the potential to inform practice in the area of supporting parents with premature babies in NICU. This paper aimed to understand maternal psychological distress of mothers with babies on a NICU and its relation to intolerance to uncertainty and formal and informal support. Results showed that formal support negatively predicted psychological distress above and beyond intolerance to uncertainty and informal social support.

Overall the paper is well structured and written with appropriate methodology.

Response: Thank you.

I have a few comments which I think will enhance the paper.

Abstract

  • I would expect to see in the abstract or main body, the dates in which the research took place. Could this be added to either the abstract or methods sections.

Response: Following the reviewer’s comment, information on dates was added to the Methods section  (see: L.189). 

Background

2)     This is very informative but very long, taking over a third of the overall paper. Would some of this be better placed in the discussion section or left out entirely. The first paragraph or two of each sections may be enough or a reduction in each section.

 Response: Following the reviewer’s comment, we shortened the Introduction section from 2193 words to 1827 words.

3)      Line 75-6 ‘The sight of their tiny infants connected to equipment by 75 tubes and wires and surrounded by medical personnel is also very disturbing’. There needs to be a reference.

 Response: Following the reviewer’s comment, the reference was added (see: L.69, 489-491).

4)      78-81 ‘Preterm mothers report feelings of 78 disappointment and frustration that they cannot perform normal parenting tasks (e.g., 79 holding, feeding, changing diapers)...’ There needs to be a reference.

 Response: This sentence was removed as part of the introduction section shortening.

Methods

5)      Methods section usually starts with an explanation of the method chosen and then goes onto the finding. This section seems to be in the wrong order.

The section on measures would be better described before providing the descriptive data, in a similar manner to your other publication or add a statement to refer to your other paper for a description of the methods. Kestler-Peleg, M., & Lavenda, O. (2022). Personal resources associated with peripartum depression among mothers of NICU 504 hospitalized preterm infants. Psychology & Health, 37, 712-730. https://doi.org/10.1080/08870446.2021.1873336

Response: Following the reviewer’s comment, the structure of the section was changed. The information regarding the study’s procedure now precedes the description of the sample (see: L.188-201).

6)      There is not enough information on the method of data collection to reproduce this study. Adding the information below would aid understanding of the process of consent and data collection.

The time period or date of the study needs to be added.

More description around the process of giving out the questionnaires would help to understand the process. When were they given out, on admission to NICU? Withing a week of admission?

Was it a paper copy, self-completed?

Response: Following the reviewer’s comments information on time period, collection procedure and the actual process of questionnaire admission was added to the Methods section (see: L.189, 191-201).

7)      Line 214, this says ‘29 Israeli mothers’. The introduction says 129. Should this be 129?

Response: Thank you for this comment, following which we have corrected the typo (see: L.190). 

8)      Line 223, the authors guidelines state the manuscript must contain ‘a statement including the project identification code, date of approval, and name of the ethics committee’. Please add these as required. 

Response: The information was added (see: L.191-193).

Results

Appropriate statistical analysis for the screening tools used and hypothesis.

Response: Thank you.

Discussion

The discussion clearly lays out the findings in relation to the hypothesis and includes limitations and practical implications.

Response: Thank you.

References

9)      There are a large number of references that are over 5 years (the time period suggested by the journal). I understand the need for the use of seminal papers and references to screening tools which are used which may be decades old. There are other references which could be updated, for example lines 432, 473, 496, 498 the reference for mothers experience of giving birth to a premature infant. There are numerous studies looking at experiences of mothers with premature babies.

e.g. The forgotten mothers of extremely preterm babies: a qualitative study. Fowler et al. 2019 or

Experiences of mothers with preterm babies at a mother and baby unit of a tertiary hospital: a descriptive phenomenological study. Lomotey et al 2019

Could these references be reviewed and updated where possible.

Response: Thank you for introducing us to these updated studies, they certainly contribute to improving the manuscript. Following the reviewer's comment, we revised the References section with more updated articles (see for example: L. 30, 55, 59, 486-488, 548-551).

Round 2

Reviewer 1 Report

Having the opportunity to review again the manuscript: I have to recognize the improvement made by the authors. Based on the recommendations, the authors addressed them properly, as well as empowered the manuscript's value. I would agree with the manuscript being published.